# Research Progress of *Ferula ferulaeoides*: A Review

**DOI:** 10.3390/molecules28083579

**Published:** 2023-04-19

**Authors:** Zhengqiong Chen, Gang Zhou, Shengjun Ma

**Affiliations:** 1College of Food Science and Pharmacy, Xinjiang Agricultural University, Urumqi 830052, China; chen18208822026@163.com; 2Key Laboratory of Ethnic Medicine and Traditional Chines, Xinjiang Uygur Autonomous Region Institute for Drug Control, Urumqi 830054, China

**Keywords:** *Ferula ferulaeoides*, chemical composition, pharmacological activity, quality control

## Abstract

*Ferula ferulaeoides* (Steud.) Korov is one of the traditional ethnic medicines in Xinjiang Uygur and Kazakh of China, which mainly contains volatile oils, terpenoids, coumarins and other chemical components. Previous work has shown that *F. ferulaeoides* exhibited insecticide, antibacterial, antitumor properties, and so on. In this paper, the chemical composition, pharmacological activity, and quality control of *F. ferulaeoides* were reviewed, and the application of *F. ferulaeoides* in the food industry was explored, so as to provide some reference for the quality evaluation of *F. ferulaeoides* and its further development and utilization.

## 1. Introduction

*Ferula* is a perennial herb belonging to the genus *Ferula* L. in the family Apiaceae, which is an important source of resin used in folk medicine [1]. There are about 180 species in the world, mainly distributed in the Mediterranean and Central Asia [2]. Among them, there are 27 species in China, 7 of which are unique [3], mainly distributed in Xinjiang, Tibet, Qinghai, Yunnan, and other provinces [4]. *Ferula* plants have a long history as a medicine. *Ferula* was utilized to treat many diseases as early as ancient Persia, so it was called “God’s food”. Nowadays, *Ferula* is still an indispensable member of Chinese herbal medicine in the international market. The medicinal material of *Ferula* is an oil–glue–resin air-dried lump, which is isolated from the genus *Ferula*. It is bitter in taste and warm in nature, and has benefits for the spleen and stomach. *Ferula* comprises three major fractions, including resin (40–64%), gum (25%) and essential oil (10–17%). The resin fraction includes ferulic acid and its esters, coumarins, sesquiterpene coumarins, and other terpenoids. The gum contains glucose, galactose, l-arabinose, rhamnose, glucuronic acid, polysaccharides, and glycoproteins, and the volatile fraction includes sulfur-containing compounds, monoterpenes, and other volatile terpenoids [5]. *Ferula ferulaeoides* (Steud.) Korov is one of the genera *Ferula*, which is often distributed in sand dunes or gravel Artemisia deserts in Mongolia, Russia, and Kazakhstan, and mainly grows in the Gobi desert in the marginal area of Junggar, Xinjiang in China [6]. *F. ferulaeoides* is widely used as a substitute for medicinal *Ferula* in Xinjiang. Its active ingredients are mainly volatile components. It has significant pharmacological activities in the treatment of insect accumulation, meat accumulation, lumps, abdominal pain, malaria, dysentery, etc. It is mostly used clinically to treat cold pain in the heart and abdomen, chronic gastroenteritis, gastric ulcers, rheumatoid arthritis, and other diseases [7].

In recent years, *F. ferulaeoides* has attracted the attention of researchers at home and abroad. It has been discovered that the main chemical components of *F. ferulaeoides* include volatile oil, resin, gum, etc. Sesquiterpenes, coumarins, and other chemical components may be the main pharmacological active components. What is more, modern pharmacological studies have shown that *F. ferulaeoides* has significant pharmacological activities such as insecticidal, bacteriostatic, anti-tumor, etc. In order to further develop and utilize this national medicinal material and fully understand its research status, this paper expounds on the research status of the chemical composition, pharmacological activity, and quality control of *F. ferulaeoides,* and looks forward to the application of *F. ferulaeoides* in the food industry, with the objective of providing some reference for the quality evaluation and in-depth development and utilization of resources.

## 2. Chemical Constituents of *F. ferulaeoides*

The chemical constituents of *F. ferulaeoides* mainly include coumarins, sesquiterpenes, and volatile oils [8]. The coumarins are one of the characteristic components of *Ferula*, and they are also the first compounds isolated from *Ferula*. The coumarins in *F. ferulaeoides* are mainly in the form of sesquiterpene derivatives, and simple coumarins are relatively rare. Most of the sesquiterpene coumarins in other *Ferula* plants are 7-hydroxy coumarins, while *F. ferulaeoides* contains mainly furan coumarins. As another characteristic component of *Ferulae*, sesquiterpenes mainly exist in the form of esters and lactone, and a few exist in the form of ketone derivatives (Table 1, Structures of sesquiterpenes are shown Figure 1, Figure 2 and Figure 3). The constituents of the volatile oil obtained from *F. ferulaeoides* have been studied more over recent years. More than 100 compounds have been separated from the volatile oil, including monoterpenes, sesquiterpenes, fatty family compounds, aromatic family compounds, and alcohol ester compounds, etc., (Table 2) [6,8,9]. At the same time, it also contains phenols or phenolic acids, phenylpropanoids, steroids and other compounds (Table 3). In addition, polysaccharides, flavonoids, and ferulic acid are also effective components of *F. ferulaeoides*, but at present, there are few studies on polysaccharides and flavonoids, and they still remain in the extraction process and content determination. Currently, ultrasonic extraction and Soxhlet extraction are the most commonly used methods to extract polysaccharides and flavonoids of *F. ferulaeoides* [10,11]. Ferulic acid is a phenolic acid, rarely exists in free form, mainly binds with oligosaccharides, polyamines, lipids, and polysaccharides, and has antibacterial and anti-inflammatory, antioxidant, antithrombotic, and other pharmacological effects [12].

## 3. Pharmacological Effects of *F. ferulaeoides*

### 3.1. Antimicrobial Activity

The emergence of bacterial resistance to different classes of antibacterial agents such as β-lactams, quinolones, and macrolides is a major problem that seriously affects human health. Therefore, during the past two decades, researchers have paid great attention to the development of antimicrobial agents, especially those of a natural origin. In addition to their efficacy, most of the natural products are non-toxic, so they can be used as a safe treatment strategy [20]. Among medicinal plants, the *Ferula* species has been identified as a rich source of antimicrobial compounds, with a variety of Ferulic plants having antibacterial effects (Table 4). The inhibitory effect of *F. ferulaeoides* on microorganisms was mainly manifested in *Staphylococcus aureus*, *Bacillus subtilis*, and *Sarcina.* Liu et al. isolated some terpenoid derivatives from *F. ferulioides* and used them in the antibacterial experiments of drug-resistant *S. aureus* strains including SA1199B (resistant to fluoroquinolones), XU212 (resistant to both tetracycline and methicillin), ATCC25923 (non-resistant), RN4220 (resistant to erythromycin), EMRSA15 (epidemic hospital MRSA), and EMRSA16 (epidemic hospital MRSA). These terpenoid derivatives showed obvious antibacterial activity [17,21]. Gao et al. carried out bacteriostatic tests on the extract and alcohol extract of *Ferula sinkiangensis, Ferula ferulaeoides*, and *Ferula sinkiangensis* leaves by the disc diffusion method. The results showed that the alcohol extracts of three kinds of *Ferula* leaves had good bacteriostatic effects on *S. aureus*, *B. subtilis*, and *Eight fold aureus,* among which *F. ferulaeoides* had the strongest inhibitory effect [22].

### 3.2. Antitumor Effect

In recent years, the treatment of various cancers with traditional Chinese medicine has attracted the extensive attention of scholars at home and abroad, and there is an urgent need to find safe and effective anti-tumor drugs from traditional Chinese medicine resources [27]. *F. ferulaeoides* is a traditional Chinese medicine in Xinjiang, and its anti-tumor research has also attracted the attention of domestic scholars. Yang et al. detected the inhibition effects of different extracts (with volatile oil, 95% ethanol extract, petroleum ether, chloroform, ethylacetate, n-butanol, and water fraction) from *F. ferulaeoides* on five types of gastric cancer cell lines (AGS, MKN-45, BGC-823, MGC-803 and SGC-7901). As a result, volatile oil, 95% ethanol extract and its petroleum ether, chloroform, and ethyl acetate fraction on five types of gastric cancer cells had different proliferation inhibition effects. Among them, the chloroform fraction had a good sensitivity to the five types of gastric cancer cell lines, with the highest sensitivity in the gastric cancer cell lines SGS-7901, and the volatile oil had a strong inhibitory effect on gastric cancer cell AGS [28]. Malignant peripheral-nerve sheath tumors (MPNSTs) are the sixth most common invasive soft tissue sarcoma, originating from the Schwann cell lineage or its precursors, with highly invasive properties in relation to surrounding peripheral nerves and there is currently no effective treatment. DAW22, a natural sesquiterpene coumarin isolated from *F. ferulaeoides* by Li et al., was found to inhibit cell proliferation and colony formation in five established human MPNST cancer cell lines, which provided strong evidence for DAW22 as a potential new alternative therapy for MPNST patients [29]. Other than that, DAW22 can induce apoptosis in C6 glioma cells occurred via the mitochondria-mediated and death-receptor pathways. It inhibited C6 glioma cell growth in a time- and concentration-dependent manner with an IC_50_ value (at 24 h) of 18.92 μM [30]. *F. ferulaeoides* has also been studied in inhibiting the activity of cervical cancer cells. Ma et al. [31] studied the anti-cervical cancer activity of four kinds of ethanol extracts from Xinjiang *Ferula* including *F. ferulaeoides*. The results showed that *F. syreitschikowi*i, *F. feurlaeoides*, *F. akitschkensis*, and *F. soongarica* could inhibit and promote the apoptosis of human cervical cancer Hela cells, and the apoptosis rates were 54.82%, 48.99%, 51.83% and 69.75%, respectively.

### 3.3. Anti-Inflammatory Effect

Researchers at home and abroad have found that *Ferula* has a clear anti-inflammatory effect [32]. The anti-inflammatory activity of *F. ferulaeoides* was studied as early as 1993. For instance, Ye et al. found that three species of *Ferula* including *F. ferulaeoides* could inhibit carrageenan-induced rat voix pedis swelling [33]. A patent in 2013 proved that ferulin A, B, C, D, and E in *F. ferulaeoides* have anti-inflammatory activity with effective doses in the range of 5–15 μ mol kg^−1^ [34]. In a study completed in 2021, the researchers established the inflammation model of lipopolysaccharide (LPS)-stimulated mouse macrophages RAW264.7 and detected the content of NO by the Griess reagent method. Moreover, they conducted the correlation analysis between the NO content and HPLC spectra of different polar extracts of *Ferula sinkiangensis* K.M Shen, *Ferula fukanensis* K.M Shen, and *Ferula. Ferulaeoides* by bivariate Pearson correlation analysis, and screened the anti-inflammatory active extracts. The results showed that the 95% ethanol extract, methylene chloride extract, and ethyl acetate extract of three kinds of *Ferula* had significant anti-inflammatory activity, and *F. ferulaeoides* had significant anti-inflammatory activity at the retention time of 86 min [35].

### 3.4. Insecticidal Activity

Insecticidal active components in plants are secondary metabolites produced in the long-term co-evolution process of plants and insects, which is relatively safe to human, livestock, crops, and the ecological environment, and insects cannot easily produce resistance to them. Therefore, it meets the requirements of people for ideal pesticides [36]. The main insecticidal effects of *F. ferulaeoides* are sesquiterpenes guaiacol and volatile oil. Liu isolated a compound with insecticidal activity, guaiacol, from the methylene chloride extract of *F. ferulaeoides* root, and tested its insecticidal activity against aphids. The results verify that the killing rate of guaiacol to aphid was almost 100%. Furthermore, guaiacol has obvious killing activity on the fourth instar armyworm, third instar cabbage moth, and the housefly [37]. *Oncomelania hupensis* (*O. hupensis*) is the unique intermediate host of Schistosoma japonicum. Studies have shown that guaiacol in *Ferula* has a killing effect. In 2007, Li et al. initially observed that *ferula* had the effect of killing *O. hupensis*, and in 2015, Fu and Zhao further proved that the mechanism of *O. hupensis* was killed mainly through guaiacol affecting its esterase and glycogen [38,39]. It was detected that the essential oil of *F.ferulaeoides*. had a strong repellent activity against the adults of the stored pest, Tribolium castaneum, and the 2nd instar larvae of Plutella maculipennis Curtis and the 10th instar larvae of yellow mealworm [40].

### 3.5. Toxic Effect

The acute toxicity of volatile oil of *F. ferulaeoides* is low. The toxicity and death of mice were observed after the mice were given an emulsion of the volatile oil of *F. ferulaeoides*. The mortality of the mice was used as the index, and the median lethal dose of the volatile oil to mice was determined by the Bliss method; the LD_50_ was 10240 mg·kg^−1^ g [41].

## 4. Research on Quality Control

### 4.1. Traits and Microstructure

*F. ferulaeoides* is characterized by granular, teardrop-like or irregular lumps, with a yellow-white to dark brown surface, light internal color, soft texture, and sticky teeth. It smells similar to celery, but is stronger than celery, with a slightly bitter and pungent taste [42]. Ding et al. [43] observed spiral or reticulate ducts, non-glandular hairs, stone cells, and sub-prism of calcium oxalate in the microstructure of *F. ferulaeoides* powder, and found a large number of resin channels of different sizes distributed in the cross-sectional microstructure of the roots, stems, and leaves of *F. ferulaeoides*, with the maximum diameter up to 10 μm. The resin canal, as its name implies, refers to the secretory tissue that secretes and synthesizes resin, mainly the secretory tube. In another study, Liu et al. investigated the microstructure and ultrastructure of the secretory ducts from the perspective of development. Their research determined that the formation model of SDs in *F. ferulaeoides* was schizogenous and pectinase contributed to SDs’ formation, while resin production was due to the activity of organelles and cytoplasm of secretory cells [44].

### 4.2. Studies on Fingerprint

Chinese medicine fingerprinting technology is an effective method for evaluating the merits of Chinese medicines, identifying authenticity, distinguishing species and ensuring their consistency and stability. In addition, it is currently the most effective means for identifying drug varieties and evaluating drug quality at home and abroad [45]. At the moment, the fingerprint applied to *F. ferulaeoides* mainly focuses on the DNA fingerprint and GC-MS fingerprint. By comparing the results of different primer increases, Miao [46] selected 15 primers of ISSR and 15 primers of RAPD to establish a DNA fingerprint profile for *F. ferulaeoides*, which provides a reliable method for scientific evaluation, effective control of herb quality, and rapid and accurate identification of similar species of the same genus. Sheng et al. [47] established GC-MS fingerprints of 44 samples of essential oils of *F. ferulaeoides* from 8 producing areas. Twelve common peaks were established by analyzing forty-four essential oil samples of *F. ferulaeoides*. The GC-MS fingerprint of in vitro anti-gastric cancer active parts from *F. ferulaeoides* was investigated and analyzed by GC-MS and principal component analysis by Wang and colleagues. The method can characterize the whole information of the chloroform extraction part of the *F. ferulaeoides* with 11 common peaks to a greater extent, and the corresponding compounds are, respectively, 3-methoxy-1,2-propanediol, D-limonene, L-borneol acetate, terpinyl acetate, 1,5,9-undecatriene, 2,6,10-trimethyl, α-cedrene, and a-bergsmotene, β-cedrene, 8-epi-γ-eudesmol, γ-eudesmol, and hinesol, which are aliphatic, aromatic, monoterpene, sesquiterpene, and their oxygenated derivatives [48]. The GC fingerprint of the volatile oil of *F. ferulaeoides* and *Ferula multicolor* was established by Luo. Two kinds of GC fingerprints for volatile oil from *ferula* have twenty common peaks in which the fingerprint similarity was more than 0.9. In addition, Luo also established two kinds of HPLC fingerprints for water extract from *ferula*, and there were thirteen common peaks in the HPLC fingerprint of *F. ferulaeoides* while there were twenty-one common peaks in the HPLC fingerprint of *Ferula* multicolor [49]. In summary, the common peaks of *Ferula ferulaeoides* are almost identified as a volatile oil, and it is necessary to strengthen the identification of other chemical constituents.

### 4.3. Content Determination

The determination of the content of *Ferulae* mainly focused on flavonoids, polysaccharides, and ferulic acid. The best extraction technology for total flavonoids is as follows: 80% ethanol, reflux extraction at 70 °C for 60 min, which could extract 29.45 mg/g of total flavonoids from 1 g of Uygur medicine *F. ferulaeoides* [10]. Ultrasonic extraction of *F. ferulaeoides* polysaccharide is better than Soxhlet extraction. The content of polysaccharide was determined by phenol-sulfuric acid colorimetry. The results show that the content of polysaccharide extracted by Soxhlet extraction and ultrasonic extraction is 2.87% and 3.22%, respectively [11]. The best extraction process of ferulic acid was the ratio of material to liquid ratio of 1:40, sodium hydroxide solution of 3%, and an extraction time of 30 min. Under these conditions, the content of ferulic acid from *F. ferulaeoides* reached 0.2998 mg/g [12]. The determination of coumarin content has also been involved; Zhu et al. established the UPLC method for the determination of DAW22 from *F. ferulaeoides* of different growth periods in Xinjiang. That indicated that the linear range of DAW22 was 6.21–124.2 ng (r = 10,000) with an average recovery of 99.81% (RSD 2.0%). The content of DAW22 in *F. ferulaeoides* growing on May 9 was the highest [50]. In general, there are few studies on the content determination of *F. ferulaeoides* and the research methods are relatively old. Therefore, modern technologies such as liquid chromatography–mass spectrometry should be used to strengthen the quantitative analysis of other chemical components, especially its characteristic components.

## 5. Prospect: Application of *F. ferulaeoides* in Food Industry

Firstly, *F. ferulaeoides* can be used as raw materials for food or food additives for food processing. There have been studies on the processing of ferulic acid into food and as a food additive. For example, some nomads of central Iran use the dried aerial parts of *F. assafoetida* L. in the preparation of their delicious local food, “Loghri”, which also contains barley, tomato or tomato paste, beans, and other vegetables. In America, different organs of *F. assafoetida* L., either in fresh or dried form, are used for cooking as even small parts of this plant can give a pungent smell to foodstuffs. It has also found many applications as a condiment and flavoring agent in chocolates, seasonings, and soft drinks [2]. Moreover, it is proved by research that felllloylphenethylami and femloyltymmine could effectively inhibit the increase in acid values, polar substances’ content, and the oxidative degradation of linoleic acid and 1inolenic acid in frying oil [51]. Secondly, the volatile oil and alcohol extract of *F. ferulaeoides* have a strong bacteriostatic effect, which can be used to make slow-release capsules or preservative agents for plastic wrap applied to food to have a bacteriostatic and preservative effect. Niazmand et al. [52] developed antimicrobial films by incorporating the hydroalcoholic extract of *Ferula asafetida* leaf and gum in the polymer matrix of LDP and investigated the antimicrobial effect of the films on different microorganisms and the capability of the produced films for extending the shelf life of the dough. The results show that as a bioactive packaging material, LDPE film containing *Ferulae* leaves and gum extract is promising, which not only improves food safety and quality, but also has good mechanical, thermal, and barrier properties. Valinezhad et al. [53] prepared a kind of composite material which called chitosan *Ferula gummosa* EO-nanocomposite (CS-FEO). Further, a transparent and flexible CS-FEO biopolymer film was prepared and characterized. The obtained results demonstrate that the prepared CS-FEO nanocomposite could be a potential candidate for food and biomedical applications as it holds the promising capability of proper interactions and advanced features. Finally, *F. ferulaeoides* can be used as a raw material for functional food development. With the improvement in people’s quality of life and the enhancement in their awareness of health care, people’s consumption of food with health care function has improved significantly. Functional food has gradually opened the market and become the future development trend of the food industry [54]. *F. ferulaeoides* contains a variety of bioactive ingredients, which is an ideal material for the development of functional food.

## 6. Conclusions

All in all, as a national medicine, the quantity of research on *F. ferulaeoides* is relatively small and not deep enough. First, the modern research of *F. ferulaeoides* mostly focuses on the studies of active ingredients such as sesquiterpenes, volatile oils, coumarins, etc., and less on active ingredients such as phenylpropanoids, phenolic acids, and other steroids. Second, the research on active ingredients mainly stays in the activity screening, mostly focusing on antibacterial, anti-tumor, anti-inflammatory, and insecticidal aspects, and there is a lack of research into its mechanisms. Third, there is a lack of drug safety evaluation in quality control research, and the identification of the fingerprint is insufficient in its depth. The existing reports are almost focused on the volatile oil, there are relatively few content determinations, and the method is more traditional. Fourth, as a kind of medicine and food homologous substance, there is almost no application research in food compared with other *Ferula* plants. In the future work, we should definitely strengthen the study of other active components of *F. ferulaeoides*, and the in-depth study of the material basis and mechanism of action. The identification of other chemical components should be added, and the quantitative analysis of other chemical components should be strengthened by using modern technologies such as liquid chromatography–mass spectrometry. In addition, a safety evaluation method for *F. ferulaeoides* should be established to provide a scientific basis for a comprehensive evaluation of the quality of medicinal materials. Eventually, we should speed up the development and utilization of *F. ferulaeoides* products, and deeply explore its development and utilization value.

## Figures and Tables

**Figure 1 molecules-28-03579-f001:**
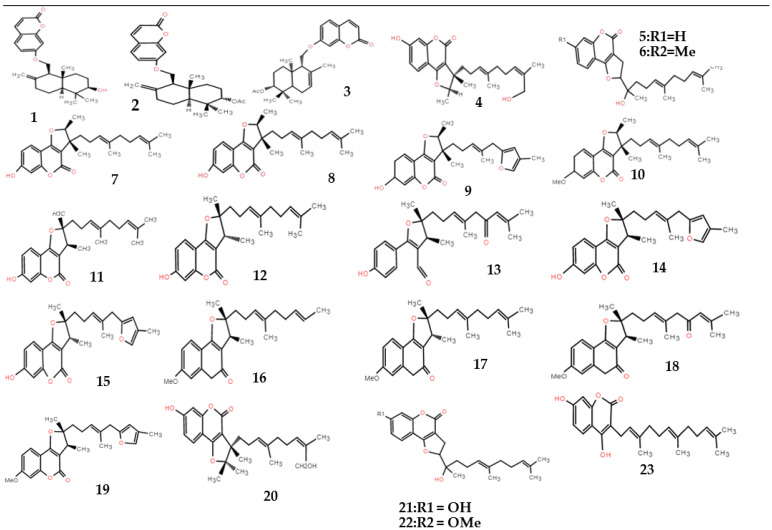
Structural diagram of sesquiterpene–coumarins of *F. ferulaeoides.*

**Figure 2 molecules-28-03579-f002:**
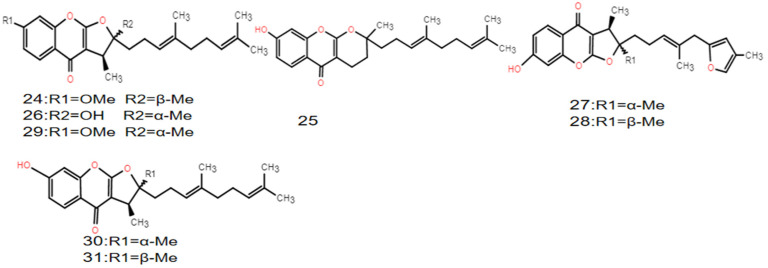
Structural diagram of sesquiterpene–heteroketone compounds of *F. ferulaeoides*.

**Figure 3 molecules-28-03579-f003:**
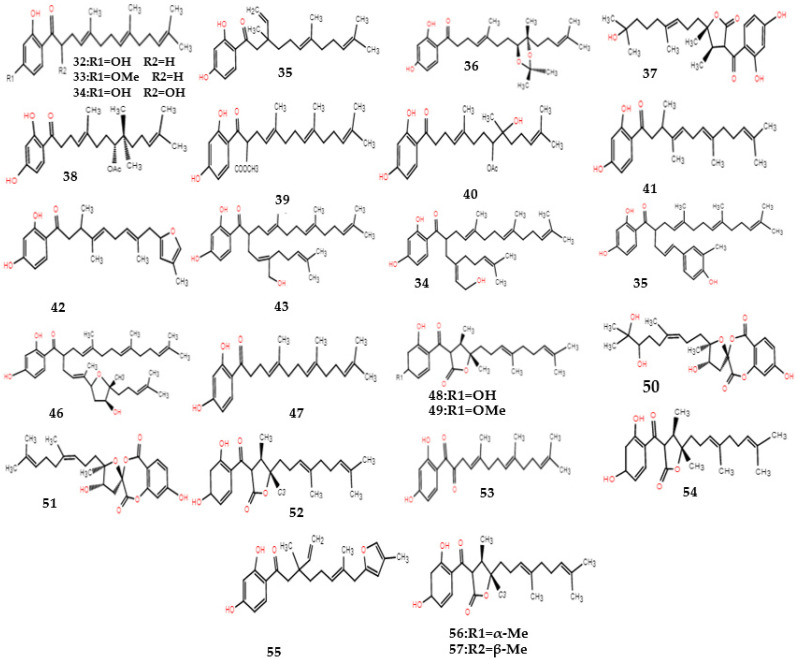
Structural diagram of sesquiterpene–phenolic compounds of *F. ferulaeoides*.

**Table 1 molecules-28-03579-t001:** Terpenoid Derivatives from Fructus *F. ferulaeoides.*

Type	No.	Compound	Formula	Molecular Weight	References
Sesquiterpene–coumarins	1	Badrakemin	C_24_H_30_O_4_	382	[13]
	2	Badrakemin acetate	C_26_H_32_O_5_	424	[13]
	3	Conferol acetates	C_26_H_32_O_5_	424	[13]
	4	Ferulin A	C_25_H_33_O_5_	413	[14]
	5	Ferulin B	C_24_H_32_O_5_	400	[15]
	6	Ferulin C	C_25_H_34_O_5_	414	[14]
	7	2,3-dihydro-7-hydroxy-2S*,3R*-dimethyl-3-[4,8-dimethyl-3(E),7-nonadienyl]-furo [3,2-c] coumarin	C_24_H_30_O_4_	382	[14]
	8	2,3-dihydro-7-hydroxy-2R*,3R*-dimethyl-3-[4,8-dimethyl-3(E),7-nonadienyl]-furo [3,2-c] coumarin	C_24_H_30_O_4_	382	[14,16]
	9	2,3-dihydro-7-hydroxy-2S*,3R*-dimethyl-3-[4-methyl-5-(4-methyl-2-furyl)-3(E)-pentenyl]-furo [3,2-c] coumarin	_C24_H_26_O_5_	394	[14]
	10	2,3-dihydro-7-methoxy-2S*,3R*-dimethyl-3-[4,8-dimethyl-3(E),7-nonadienyl]-furo [3,2-c] coumarin	C_25_H_32_O_4_	396	[15]
	11	2,3-dihydro-7-hydroxy-2S*,3R*-dimethyl-2-[4,8-dimethyl-3(E),7-nonadienyl]-furo [3,2-c] coumarin	C_24_H_30_O_4_	382	[15]
	12	2,3-dihydro-7-hydroxy-2R*,3R*-dimethyl-2-[4,8-dimethyl-3(E),7-nonadienyl]-furo [3,2-c] coumarin	C_24_H_30_O_4_	382	[15]
	13	2,3-dihydro-7-hydroxy-2S*,3R*-dimethyl-2-[4,8-dimethyl-3(E),7-nonadien-6-onyl]-furo [3,2-c] coumarin	C_25_H_32_O_4_	396	[15]
	14	2,3-dihydro-7-hydroxy-2S*,3R*-dimethyl-2-[4-methyl-5-(4-methyl-2-furyl)-3(E)-pentenyl]-furo [3,2-c] coumarin	C_24_H_26_O_5_	394	[14]
	15	2,3-dihydro-7-hydroxy-2R*,3R*-dimethyl-2-[4-methyl-5-(4-methyl-2-furyl)-3(E)-pentenyl]-furo [3,2-c] coumarin	C_24_H_26_O_5_	394	[14]
	16	2,3-dihydro-7-methoxy-2S*,3R*-dimethyl-2-[4,8-dimethyl-3(E),7-nonadienyl]-furo [3,2-c] coumarin	C_25_H_32_O_4_	396	[15]
	17	2,3-dihydro-7-methoxy-2R*,3R*-dimethyl-2-[4,8-dimethyl-3(E),7-nonadienyl]-furo [3,2-c] coumarin	C_25_H_32_O_4_	396	[15]
	18	2,3-dihydro-7-methoxy-2S*,3R*-dimethyl-2-[4,8-dimethyl-3(E),7-nonadien-6-onyl]-furo [3,2-c] coumarin	C_25_H_30_O_5_	410	[15]
	19	2,3-dihydro-7-methoxy-2S*,3R*-dimethyl-2-[4-methyl-5-(4-methyl-2-furyl)-3(E)-pentenyl]-furo [3,2-c] coumarin	C_25_H_28_O_5_	408	[15]
	20	(trans)-2,3-dimethyl-3-[9-hydroxymethyl-4-methyl-3E,7Z-nonadienyl]-7-hydroxy-2(3H)-furo [3,2-c] coumarin	C_24_H_31_O_5_	399	[15]
	21	7-hydroxy-2-[1-hydroxy-1,5,9-trimethyl-4E,8-decadienyl]-2(3H)-furo [3,2-c] coumarin	C_24_H_31_O_5_	399	[15]
	22	2-[1-hydroxy-1,5,9-trimethyl-4E,8-decadienyl]-7-methoxy-2(3H)-furo [3,2-c] coumarin	C_25_H_33_O_5_	408	[15]
	23	4,7-dihydroxy-3-[3,7,11-trimethyl-2(E),6(E),10-dodecatrienyl] coumarin	C_24_H_30_O_4_	382	[14]
Sesquiterpene–heteroketone compounds	24	Ferulin D	C_25_H_32_O_4_	396	[15]
	25	Ferulin E	C_24_H_31_O_4_	383	[14]
	26	2,3-dihydro-7-hydroxy-2S*,3R*-dimethyl-2-[4,8-dimethyl-3(E),7-nonadienyl]-furo [3,2-b] chromone	C_24_H_30_O_4_	382	[14]
	27	2,3-dihydro-7-hydroxy-2S*,3R*-dimethyl-2-[4-methyl-5-(4-methyl-2-furyl)-3(E),7-pentenyl]-furo [2,3-b] chromone	C_24_H_26_O_5_	394	[15]
	28	2,3-dihydro-7-hydroxy-2R*,3R*-dimethyl-2-[4-methyl-5-(4-methyl-2-furyl)-3(E),7-pentenyl]-furo [2,3-b] chromone	C_24_H_26_O_5_	394	[15]
	29	4-hydro-7-hydroxy-2-methyl-2-[4,8-dimethyl-3E,7-nonadienyl]-2(3H)-pyro [2,3-b] chromone	C_24_H_31_O_4_	383	[14]
	30	2,3-dihydro-7-hydroxy-2S*,3R*-dimethyl-2-[4,8-dimethyl-3(E),7-nonadienyl]-furo [3,2-b] chromone	C_24_H_30_O_4_	382	[15]
	31	2,3-dihydro-7-hydroxy-2R*,3R*-dimethyl-2-[4,8-dimethyl-3(E),7-nonadienyl]-furo[3,2-b] chromone	C_24_H_30_O_4_	382	[15]
Sesquiterpene–phenolic compounds	32	Dshamirone (secoammoresinol)	C_23_H_33_O_3_	356	[15]
	33	(4E,8E)-1-(2,4-dihydroxyphenyl)-5,9,13-trimethyl-tetradeca-4,8,12-trien-1-one	C_23_H_32_O_3_	356	[14]
	34	(4E,8E)-1-(2,4-dihydroxyphenyl)-2-methoxycar-bonyl-5,9,13-trimethyltetradeca-4,8,12-trien-1-one	C_25_H_34_O_5_	414	[14]
	35	1-(2,4-dihydroxyphenyl)-3,7,11-trimethyl-3-vinyl-6(E),10-dodecadiene-1-one	C_23_H_32_O_3_	356	[15]
	36	8,9-oxoisopropanyldshami-rone	C_22_H_30_O_5_	374	[17]
	37	Ferulaeolactone A	C_24_H_34_O_6_	418	[17]
	38	8-acetoxy-9-hydroxydshmirone	C_24_H_34_O_5_	402	[17]
	39	Ferulaeone A	C_25_H_33_O_5_	413	[15]
	40	Ferulaeone B	C_25_H_37_O_6_	433	[15]
	41	Ferulaeone C	C_23_H_31_O_3_	355	[15]
	42	Ferulaeone D	C_23_H_28_O_4_	369	[15]
	43	Ferulaeone E	C_33_H_49_O_4_	509	[15]
	44	Ferulaeone F	C_33_H_49_O_4_	509	[15]
	45	Ferulaeone G	C_33_H_43_O_5_	519	[15]
	46	Ferulaeone H	C_38_H_55_O_5_	591	[15]
	47	3-(2,4-dihydroxybenzoyl)-4S*,5R*-dimethyl-5-[4,8-dimethyl-3(E),7(E)-nonadien-1-yl] tetrahydro-2-furanone	C_24_H_32_O_5_	400	[15]
	48	3-(2-hydroxyl-4-methoxybenzoyl)-4S*,5R*-dimethyl-5-[4,8-dimethyl-3(E),7(E)-nonadien-1-yl] tetrahydro-2-furanone	C_25_H_34_O_5_	414	[15]
	49	3-(2,4-dihydroxybenzoyl)-4R*,5R*-dimethyl-5-[4,8-dimethyl-3(E),7(E)-nonadien-1-yl] tetra-hydro-2-furanone	C_24_H_32_O_5_	400	[15]
	50	Ferulactone A	C_24_H_32_O_9_	463	[18]
	51	Ferulactone B	C_24_H_30_O_7_	429	[18]
	52	3S*-(2,4-dihydroxybenzoyl)-4R*,5R*-dimethyl-5-(4,8-dimethyl-3(E),7(E)-nonadien-1-yl) tetrahydro-2-furanone	C_24_H_32_O_5_	400	[19]
	53	1-(2,4-dihydroxyphenyl)-2-hydroxy-5,9,13-trimethy1-4(E),8(E),12-tetradecatrien-1-one	C_23_H_32_O_4_	372	[19]
	54	1-(2,4-dihydroxyphenyl)-3,7,11-trimethyl-3-vinyl-6(E),10-dodecadiene-1,9-dione	C_23_H_30_O_4_	370	[19]
	55	1-(2,4-dihydroxyphenyl)-3,7-dimethyl-3-vinyl-8-(4-methyl-2-furyl)-6(E)-octen-l-one	C_23_H_28_O_4_	368	[15]
	56	3S*-(2,4-dihydroxybenzoyl)-4R*,5R*-dimethyl-5-[4-methyl-5-(4-methyl-2-furyl)-3(E)-penten-1-yl] tetrahydro-2-furanone	C_24_H_28_O_6_	412	[15]
	57	3S*-(2,4-dihydroxybenzoyl)-4R*,5S*-dimethyl-5-[4-methyl-5-(4-methyl-2-furyl)-3(E)-penten-l-yl] tetrahydro-2-furanone	C_24_H_28_O_6_	421	[15]

**Table 2 molecules-28-03579-t002:** Main chemical compositions in volatile oil of *F. ferulaeoides.*

Type	Compound	References
Monoterpenoids	α-pinene, β-pinene, camphene, δ-3-carene, limonene, D-limonene, L-limonene, α-phellandrene, β-thujene, γ-terpinene, ocimene, β-ocimene, terpinolene, α-terpinolene, myrcene, β-myrcene, sabinene, fenchene, fenchol, borneol, (R)-camphor, citronellol, P-cymen-8-ol, α-terpined, 1,7,7-trimethyl-exo-bicyclo[2.2.1]heptan-2-ol, tanacetone, Z-3-pnen-2-ol, isogeraniol), 3-methoxy-p-cymene, dihydrocarvyl aetate, thymylether methyl, 2-camphanol acetate, exobornyl acetate, α-fenchyl acetate, [1,3,6-octatriene,3,7-dimethyl-,(Z)-], 3-isopropylidene-5-methyl-hex-4-en-2-one)	[8]
Sesquiterpene	Farnesene, α-guaiene, α-farnesene, β-farnesene, α-curcumene, β-curcumene, α-elemene, β-elemene, valencene, (−)-alloaromadendrene, trans-caryophyllene, aristolene, α-gurjunene, α-cedrene, β-cedrene, hujopsenet, α-himachalene, β-himachalene, α-bulnesene, isoledene, (Z,E)-α-farnesene, 1.2,3,4,4A,5,6,8A-octa-hydro-naphthalene, bergamotene, γ-selinene, α-bergamotene, zingiberene, isocaryophyllene, ledene, β-bisabolene, τ-guaiene, guaiol, α-guaiol, δ-guaiol, bulnesol, 10-O-γ-eudesmol, α-eudesmol, τ-eudesmol, β-eudesmol, elemiol, cedrol, hinesol, agarospirol, ginsenol, calareneexoide, eudesm-7(11)-en-4-ol, β-bisabolol, nerolidol, trans-nerolidol, 1,2- propanediol,3-methoxy-, 2,6,6-trimethyl-1- methylen-cyclohex-2-ene, 5,9-undecadien- 2-one,6,10-dimethyl-,(Z)-, 1,1-dimethyl-2,4-di(1-propenyl)cyclohexane, (Z)-2,6,10- trimethyl-1,5,9-undecatriee, 5-β-H,7-β, 10-α-selina-4(19),11-diethy, 4-(1E)-1,3-butadien-1-yl-3,5,5-trimethyl]	[8]
Aromatic	Toluene, resorcinol, O-cymene, benzene, 2-methoxy-4-vinylphenol, 1-methyl-4-(1-methylethyl) benzene, 4-(1-methylethyl)-benzene-methanol, 1-1,5-dimethyl-4- hexenyl-4-methyl benzene	[8]
Alcohol esters	cyclopentanol,2-methyl acetate, neryl acetate, butane-2,3-diol, (2S,3S)-(+)-2,3-butane diol, 4,8-dimethyl-3,7-nonadien-2-ol, 4-terpinyl acetate, α-terpinyl acetate, ethyl palmitate, ethyl linoleate	[8]

**Table 3 molecules-28-03579-t003:** Other chemical constituents of *F. ferulaeoides.*

Type	No.	Compound	Formula	Molecular Weight	References
Phenylpropanoids	1	myristicin	C_11_H_12_O_3_	192	[19]
Phenols, phenolic acids	2	2,4-dihydroxyacetophenone	C_8_H_8_O_3_	152	[19]
	3	2-hydroxy-4-methoxyacetophenone	C_9_H_10_O_3_	166	[19]
	4	2,4-dihydroxybenzoic acid	C_7_H_6_O_4_	154	[19]
	5	2,4-dihydroxy-α-oxobenzeneacetic acid	C_8_H_8_O_4_	168	[19]
	6	β-resorcylic acid	C_7_H_6_O_4_	154	[19]
	7	Methoxyresorcylic acid	C_8_H_8_O_4_	168	[19]
	8	Umbelliferone	C_9_H_6_O_3_	162	[6]
	9	Lehmannmlone	C_24_H_30_O_4_	382	[6]
	10	methyl 2,4-dihydroxybenzoate	C_8_H_8_O_4_	168	[6]
	11	ethyl 2,4-dihydroxybenzoate	C_9_H_10_O_4_	182	[6]
Steroids	12	β-sitosterol	C_29_H_50_O	414	[19]
	13	Daucossterol	C_35_H_60_O_6_	576	[6]

**Table 4 molecules-28-03579-t004:** Antibacterial Effect of *Ferula* Plants.

*Ferula* Plants	Antibacterial Ingredients	Microorganism	References
*F. lycia*	essential oil	*Haemophilus influenza T*	[23]
*F. glauca*	essential oil	*Streptococcus mutans*, *Enterococcus faecalis*, *Escherichia coli*	[23]
*F. heuffelii*	essential oil	*Micrococcus luteus*, *Staphylococcus epidermidis*, *B. subtilis,Micrococcus flavus*	[23]
*F. assafoetida*	organic extractswater extracts	*coli*, *S. aureus*, *E. faecalis*, *Shigella flexneri*, *Klebsiella pneumonia*	[21,23]
*F. gummosa*	oleo-resin	E. *coli*, *P. aeruginosa*, *S. aureus*, *Salmonella enteritidis, Listeria monocytogenes*	[23]
	essential oil	*C. albicans*, *S. epidermidis*, *S. aureus*, *E. coli*, *B. cereus*, *E. faecalis*, *P. aeruginosa*	
*F. communis*	petroleum ether extract	*S. aureus*, *B. subtilis*, *Streptococcus durans,* E. faecalis	[23]
*F. szovitsiana*	essential oil	*B. subtilis*	[23]
*F. hermonis*	essential oil	*S. typhi*, *P. aeruginosa*, *E. coli*, *S. aureus*, *S. fecalis*	[23]
*F. vesceritensis*	essential oil	*E. coli*, *K. pneumonia*, *S. aureus, P. aeruginosa*	[23]
*F. kuhistanica*	essential oil	*S. aureus*	[23]
*F. ferulaeoides*	sesquiterpene derivatives	Drug-resistant *S. aureus*	[14][22]
	alcohol extract	*S. aureus*, *B. subtilis*, *Sarcina*	
*F. elaeochytris*	essential oil	*S. aureus*	[24]
*F.pseudalliacea*	sesquiterpene coumarins	*S. aureus*, *Enterococcus faecium*, *B. cereus*, *E. coli*, *P. aeruginosa*	[23]
*F. tunetana*	essential oil	*Salmonella typhimurium*, *S. epidermidis*, *Micrococcus luteus*, *B. Cereus*, *B. subtilis*	[25]
*F. tingitana* L.	essential oil	*Bacillus subtilis*, *Neisseria gonorrhoeae*	[26]

## Data Availability

Data will be provided upon request.

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
