# Peer review of "Research Progress of Ferula ferulaeoides: A Review"

_molecules, 2023, doi:10.3390/molecules28083579_

Round 1

Reviewer 1 Report

Name of family:Apiaceae

Name of species by name of othuers

Reviewer 2 Report

The article provides an overview of Ferula, a perennial herb that is an important source of resin used in folk medicine. The genus Ferula consists of about 180 species worldwide, and 27 species in China, 7 of which are unique. The article explains that Ferula plants have a long history as medicine, and the medicinal material of Ferula is an oil glue resin air-dried lump, which is bitter in taste and warm in nature. Ferula comprises three major fractions, including resin, gum, and essential oil, which have different chemical compositions and pharmacological activities. F. ferulaeoides is one of the Ferula species that is mainly distributed in the Gobi desert in the marginal area of Junggar, Xinjiang in China. It has significant pharmacological activities in the treatment of various diseases such as insect accumulation, meat accumulation, lumps, abdominal pain, malaria, dysentery, and so on. The article also discusses the research status of chemical composition, pharmacological activity, and quality control of F. ferulaeoides, and looks forward to its potential application in the food industry.
I have 4 questions for authors:
1, What are the three major fractions of Ferula, and what are their chemical compositions and pharmacological activities?
2, What are the main pharmacological activities of F. ferulaeoides, and what diseases can it treat?
3, Why is there a lack of research on the mechanisms of F. ferulaeoides's pharmacological activities?
4, What is the potential application of F. ferulaeoides in the food industry, and how can it be utilized in this context?

Reviewer 3 Report

There are not many scientific works concerning F. Ferulaeoides, despite this the authors have left out the evaluation of the works that have taken into consideration the composition of the essential oil of the flowers, which instead could give greater importance to the work. I therefore recommend taking this aspect into consideration, so  the work can be integrated with more information on the composition of the essential oil of ferula flowers

Extensive editing of the English language and style is required.

Check text for typos and repetitions 

for example:

line 25-27:  there is a repetition

line 263-268: there is a repetition 

Round 2

Reviewer 3 Report

thanks for the changes you made to the article